# Vulnerability in maternal, new-born, and child health in low- and middle-income countries: Findings from a scoping review

Olusesan Ayodeji Makinde[1,2]* , Olalekan A. Uthman[3], Ifeanyi C. Mgbachi[1], Nchelem Kokomma Ichegbo[1,2], Fatima Abdulaziz Sule[1], Emmanuel O. Olamijuwon[1], Babasola O. Okusanya[4]

1 Department of Research and Development, Viable Helpers Development Organization, Abuja, Nigeria,
2 Department of Research and Development, Viable Knowledge Masters, Abuja, Nigeria, 3 Department of Global Health, University of Warwick, Coventry, United Kingdom, 4 Department of Obstetrics and Gynaecology College of Medicine, University of Lagos, Lagos, Nigeria

☯ These authors contributed equally to this work.
* sesmak@gmail.com

## Abstract

### Objectives

To identify and synthesise prevailing definitions and indices of vulnerability in maternal, new-born and child health (MNCH) research and health programs in low- and middle-income countries.

### Design and setting

Scoping review using Arksey and O'Malley's framework and a Delphi survey for consensus building.

### Participants

Mothers, new-borns, and children living in low- and middle-income countries were selected as participants.

### Outcomes

Vulnerability as defined by the authors was deduced from the studies.

### Results

A total of 61 studies were included in this scoping review. Of this, 22 were publications on vulnerability in the context of maternal health and 40 were on new-born and child health. Definitions used in included studies can be broadly categorised into three domains: biological, socioeconomic, and environmental. Eleven studies defined vulnerability in the context of maternal health, five reported on the scales used to measure vulnerability in maternal health and only one study used a validated scale. Of the 40 included studies on vulnerability in child health, 19 defined vulnerability in the context of new-born and/or child health, 15

**Data Availability Statement:** All relevant data are within the paper and its Supporting Information files.

**Funding:** This work was supported in whole by the Bill & Melinda Gates Foundation [INV-015806]. The funders had no role in study design, data collection and analysis, decision to publish, or preparation of the manuscript.

**Competing interests:** The authors have declared that no competing interests exist.

reported on the scales used to measure vulnerability in child health and nine reported on childhood vulnerability indices. As it was difficult to synthesise the definitions, their keywords were extracted to generate new candidate definitions for vulnerability in MNCH.

## Conclusion

Included studies paid greater attention to new-born/ child vulnerability than maternal vulnerability, with authors defining the terms differently. A definition which helps in improving the description of vulnerability in MNCH across various programs and researchers was arrived at. This will further help in streamlining research and interventions which can influence the design of high impact MNCH programs.

## Scoping review registration

The protocol for this review was registered in the open science framework at the registered address (https://osf.io/jt6nr).

## Introduction

Vulnerability is an important concept in public health research and policy. Recently, there has been a rise in the interest in the vulnerabilities of special populations like sex workers, people living with HIV, migrant women, and orphans [1, 2]. Among these populations, there is gradience of vulnerabilities, with associated risks from one population to another. Vulnerability among children, especially in developing countries, is exacerbated by exposure to factors like poverty, lack of psychosocial and emotional support, and family problems [3]. In sub-Saharan Africa (SSA), orphans and vulnerable children (OVCs) affected by human immunodeficiency virus infection face numerous challenges to their mental, physical, and social well-being [4]. With their education affected, they are likely to live in abject poverty. For instance, in Nigeria, it is estimated that approximately 95% of OVCs do not receive the required social, emotional, or medical help needed to reduce their vulnerability [5].

Alongside the interest in vulnerabilities in special populations, there is an emerging focus on vulnerability in the context of maternal, new-born, and child health (MNCH), with efforts targeted at bridging the gap in implementation science and policy redirection in low- and middle-income countries (LMICs) [6]. This gap persists despite some progress in MNCH programming and outcomes over the past decade [7, 8]. Maternal and childhood mortality remain key health challenges in several low-and middle-income regions [9–12]. A high childhood mortality rate, particularly in the first month of life, still exists despite increasing health interventions in several regions. Even within countries, significant geographic variation in vulnerabilities in maternal and newborn health outcomes manifest as high and low mortality distribution and this requires further country level analysis for planning [13, 14]. The SSA region recorded the highest neonatal mortality rate of 27 per 1000 live-births in 2019 [15]. To identify vulnerability-mitigating factors and define how programs in MNCH can further implement targeted interventions to address persistent shortcomings, the objective of this study was to conduct a scoping review to identify and synthesise prevailing definitions and indices of vulnerability in MNCH research and health programs in low- and middle-income countries.

## Methods

Arksey and O'Malley's framework was used to conduct the scoping review to investigate the definitions and coverage of vulnerability in MNCH domains in LMICs [16, 17]. The review followed five stages: (1) identifying the research question, (2) identifying the relevant studies, (3) selecting the studies, (4) charting data, and (5) collating, summarising, and reporting results.

### Protocol and registration

The protocol for this review was registered in the open science framework [18].

### Eligibility criteria

We included studies that met the following inclusion criteria:

**Types of study.**   Peer-reviewed publications and programmatic reports that discussed vulnerability and vulnerable populations in the context of MNCH, including those in puerperium, were included.

**Types of population.**   Studies that focused on women (pregnant and lactating) and children ($< 5$ years), regardless of the gender of the child, were included.

**Types of setting.**   Any study conducted in low- and middle-income settings as classified by the World Bank Country and Lending groups [19].

### Information sources and search

We formulated a comprehensive and exhaustive search strategy (see S1 Fig) to identify all relevant studies regardless of language or publication status. **The initial search was conducted on 15 January 2021 and updated on 26 August 2022.**

Electronic databases: Databases like Medline via Ovid databases, Embase, Scopus, and Web of Science were searched using appropriate keywords. A wide search was conducted to include high-quality literature beyond the traditional sources outlined above, including advanced Google search, review of reports, and technical papers from multilateral and bilateral organisations such as the United States Agency for International Development (USAID), United Nations Children's Emergency Fund (UNICEF), Public Health England, and World Health Organization (WHO). We also searched Ovid Northern Light Life Sciences Conference Abstracts, and reference lists of all studies and reviews identified by the above methods [20].

The search strategy was structured around three blocks focusing on: (1) population (MNCH, health outcomes, healthcare utilisation, and social capital), (2) exposure (vulnerability), and (3) setting (low- and middle-income and resource-limited settings). Critical keywords and thesaurus heading terms were initially tailored to Medline and Embase searches and then adapted for other sources as necessary. S1 Fig shows the full search strategies for Medline and Embase.

### Selection of sources of evidence

To reduce the workload of screening the result from the highly sensitive search, we developed a bespoke machine learning classifier/algorithm to identify potentially relevant studies. We trained the Bidirectional Encoder Representations from Transformers (BERT) model and validated it to be a high-performance algorithm comparable to human screening, that is, desired recall (also known as sensitivity) of at least 95% [21, 22]. The training and validation sets were chosen at random from the search results across all databases. From this algorithm, the machine-made predictions (including or excluding) based on other titles and abstracts that

were included in the list. We used the Local Interpretable Model-Agnostic Explanations (LIME) to explain the reasons behind the predictions in an interpretable manner. We used the Covidence Systematic Review Software to manage the search outputs and screening of eligible studies (https://www.covidence.org/). For flexibility, we used Airtable (https://airtable.com/) to manage the data extraction. Two review authors independently screened the results of the literature search for potentially relevant studies and obtained full reports for further assessment. They independently applied the inclusion criteria to the full-text reports using the eligibility criteria and scrutinised publications to ensure that each study was included in the review only once. Disagreements were resolved through a consensus within the review team and by contacting a third author in case of a disagreement between the two authors on study eligibility. The Covidence software used permitted multiple team members to work simultaneously as either first or second reviewer. All co-authors contributed to the screening and identification of the articles. We also assessed and analysed the excluded studies and the reasons for their exclusion.

## Data items

Using a template designed in Airtable, two reviewers independently extracted the data from the included studies. We extracted data on the following: study identification, author(s), publication status, study period, details of the study (study design or type, country and location of the study), context or setting in which the study was conducted or reported, characteristics of participants, and details and definition of the vulnerability reported in the study. Furthermore, we extracted information on the vulnerability of MNCH concerning the COVID-19 pandemic to understand how the pandemic influenced the vulnerability of women and children. Additionally, we extracted information that shed light on the gender responsiveness of the studies, such as studies targeted at equity or male engagement in MNCH. All disagreements were resolved through discussions between all review authors.

## Synthesis of results

Charted information was exported into Microsoft (MS) Excel for additional coding and data analysis. Temporal patterns were displayed by plotting the yearly number of published literatures for each year. Descriptive statistics (frequency and percentage) of country affiliation, language of publication, publication type, and institutional affiliation of the authors were also calculated. The R statistical package was used to generate additional charts and world maps which showed the place where the studies were conducted.

The definitions of vulnerability used by the authors were also extracted and analysed. The number of times the definitions were used by different authors was noted. The candidate definitions of vulnerability in the context of the MNCH were then generated. In generating definitions for vulnerability in MNCH, we appraised definitions used by the authors of the included studies and extracted key constructs and words. The extracted constructs and terms were pooled together to determine prominent constructs and words by word clouding at https://tagcrowd.com [23]. This helped in including commonly used constructs and words in several combinations to create candidate definitions of vulnerability in MNCH. Definitions generated through these processes were presented to experts with previous publications on vulnerability in MNCH to vote on or make suggestions.

## Patient and public involvement statement

Patients were not involved in the conceptualization or conduct of this study.

## Results

**Study selection and characteristics.** [Fig 1] shows the study selection flowchart. A literature search yielded 88,842 citations. To reduce the workload of screening the search results, we developed a bespoke classifier/algorithm to identify potentially relevant studies. We annotated a random sample of 2,500 citations, with only 133 (5%) articles tagged as potentially relevant (included) and 2,367 as irrelevant (excluded) studies (95%). The BERT bespoke classifier showed satisfactory results, with 0.97 precision, 0.96 recall, 0.80 specificity, 0.96, accuracy and 0.96 F1-measure. Thereafter, the bespoke classifier was applied to screen all other unseen citations identified from our searches. Only 3,357 citations were labelled by the classifier as potentially relevant for the review. After reviewing the titles and abstracts of the 3,357 citations, 100 articles were selected for full-text screening. Of these, 49 did not meet the inclusion criteria and were subsequently excluded, because of wrong population (n = 25), wrong outcomes

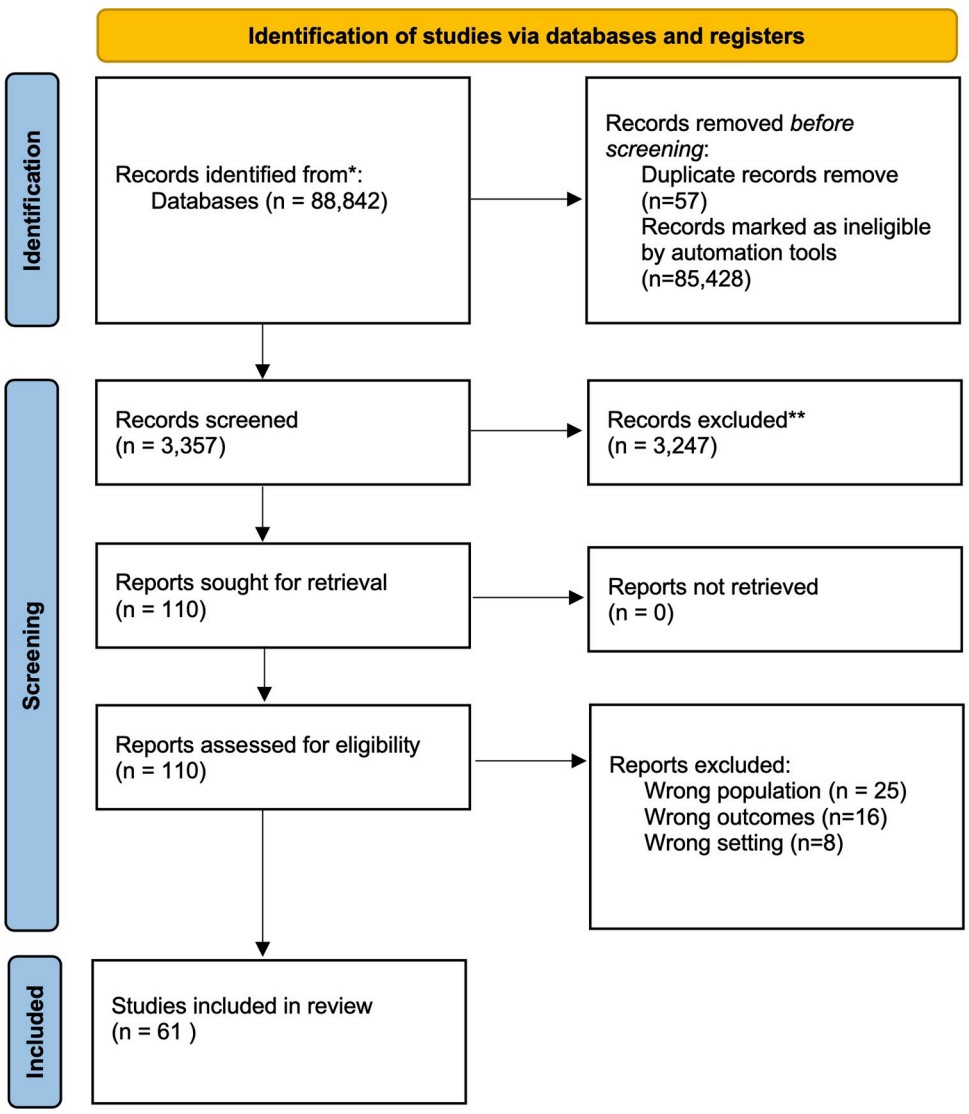

**Fig 1. PRISMA study selection process flow chart.** * Databases like Medline via Ovid databases, Embase, Scopus, and Web of Science were searched using appropriate keywords. ** Records excluded after manual screening.

reported (n = 16) and conducted in high-income countries (n = 8). Sixty-one studies met the inclusion criteria and were thus included in the scoping review. **Tables 1 and 2** present the characteristics of the included studies.

## Vulnerabilities in maternal health

Of the included studies, 22 (34%) were publications on vulnerability in the context of maternal health; most studies focused on a single country, while two focused on multiple countries. The studies were conducted across 15 countries, four were from Brazil, and two from India, Pakistan and Chile [24–33].

Eleven studies were quantitative and observational in design [24, 27–31, 34–38]. six were qualitative research [26, 32, 39–42], four used mixed-methods design [25, 33, 43, 44], and one was a randomised controlled trial [45]. The details are presented in **Table 1** and **Fig 2**.

The reported objectives varied across the studies. Arcos et al. examined the incidence of social vulnerability and its determinants, starting during the gestation period [33]. Storeng et al. examined the concept of vulnerability in the context of maternal morbidity and mortality in Burkina Faso [41], Ferreira et al. presented a spatial analysis of the social vulnerability of teenage pregnancy, through the geoprocessing of birth and death data, existing in the Ministry of Health's databases in Brazil, in order to subsidise actions and strategies in the intersectoral management process based on the problematisation of spatial analysis in the neighbourhood areas [25]. Prates and colleagues examined whether experiences of using contraceptives among poor women relate to vulnerability which is either a generator or an enhancer [26].

**Definition of vulnerability in the context of maternal health.** Fourteen studies defined vulnerability in the context of maternal health (see **Table 2**). The definitions used can be broadly categorised into three groups: biological, socioeconomic, and environmental. Some definitions had elements of two or three categories.

*Biological.* Two studies used biological vulnerability in the context of maternal health: risk of comorbid anxiety and depression in pregnancy [31], and HIV positivity [28].

*Socioeconomic.* Seven studies used financial vulnerability in the context of maternal health. These studies explored women's socioeconomic conditions, such as monthly income per capita of the household municipality and context of limited social and economic resources needed for protection against risks or mitigation of resultant social and economic consequences [24–26, 32, 38, 39, 43]. Studies explored the course of vulnerability; social threats to successful transfer, barriers to timely intrapartum care and reparative interventions [43] and multiparity [having many children] in the context of poverty, especially in a context where gender inequality undermines the prospects of young girls and women [26].

*Combined definitions: Socioeconomic, biological, or environmental.* Three studies used a combination of biological, socioeconomic, and environmental vulnerabilities in the context of maternal health. Arcos and colleagues stated that 'vulnerability is related to the biological and psychological characteristics of people, social and environmental conditions, the life cycle, the structure and functionality of the family and the territory where they live, because poverty is concentrated in territorial neighbourhood units, generating spaces of vulnerability and social exclusion that imply fragility, threat and susceptibility to health damage' [33].

Den Hollander and colleagues defined 'vulnerability as the likelihood of being wronged, i.e., being denied adequate satisfaction of certain legitimate claims to physical integrity, autonomy, freedom, social provision, impartial quality of government, social bases of self-respect or communal belonging' [46].

Theophilo and colleagues [27], categorised vulnerability into the following:

**Table 1. Characteristics of included studies in maternal health.**

| Author & year | Country | Research method | Study design | Study objectives |
|---|---|---|---|---|
| Arcos 2011 | Chile | Mixed methods | Observational study | To determine the incidence of social vulnerability and its determinants among women, starting at the gestation period. |
| Arps 2012 | Honduras | Quantitative | Observational study | To examine social and economic factors associated with maternal mortality. |
| Brahme 2019 | India | Quantitative | Observational study | The aim of the study is to understand and explain the trends in HIV prevalence among the ANC attendees of different age groups. |
| Dias 2020 | Brazil | Quantitative | Observational study | The study assessed sociodemographic variables, psychiatric parameters and thiamine and its derivative in women's blood in a rural, low-income community in Brazil. |
| Ferreira 2012 | Brazil | Mixed methods | Observational study | The aim of the study was to present a spatial analysis of the social vulnerability of teenage pregnancy, through the geoprocessing of birth and death data, existing in the ministry of health's databases, in order to subsidise actions and strategies in the intersectoral management process based on the problematisation of spatial analysis in the neighbourhood areas. |
| Ganeshan 2014 | Malaysia | Quantitative | Observational study | The objective of this study is to understand the age specific physical complications of adolescent pregnancies. Although the social implications are more dramatic, it is less easily measured. This vital step will play a greater role as we endeavour towards achieving the Millennium Development Goals. |
| *Kaye 2014 | Uganda | Qualitative | Observational study | To gain an understanding of how obstetric complications affect the lives and livelihoods of survivors. |
| Kouassi 2012 | Côte d'Ivoire | Quantitative | Observational study | To assess awareness of the pandemic and awareness and acceptance of A(H1N1) vaccine. |
| Lavender, 2020 | Tanzania & Zambia | Mixed methods | - | To gain a comprehensive understanding of the complexities surrounding intrapartum referrals in Tanzania and Zambia. |
| *McNaughton Reyes 2020 | South Africa | Quantitative | Randomised clinical trial | To identify factors that condition (i.e., buffer or exacerbate) the impact of exposure to violence inflicted by intimate partner on postpartum emotional distress among South African women. |
| Muñoz 2013 | Chile | Qualitative | - | To understand the future expectations and experience of vulnerable mothers from pregnancy to their child's early years. |
| Pourette 2012 | Haiti | Qualitative | Observational study | The objective of this paper was to show, based on the analysis of the trajectories of Haitian women living with HIV in Guadeloupe, the way in which these multiple and interconnected issues permeate these women's trajectories, particularly in the reproductive field. |
| Prates 2008 | Brazil | Qualitative | Observational study | The objective of this research was to problematise the contraceptive experiences of poor women as it relates to vulnerability which is either a generator or an enhancer. |
| Premji 2020 | Pakistan | Quantitative | Observational study | The aim of the study was to explore the prevalence and patterns of comorbid antenatal anxiety and depressive symptoms; to understand the risk factors for comorbid anxiety and depressive symptoms. |
| Santhakumar 2020 | India | Quantitative | Observational study | To analyse the demographics of HIV-positive pregnant mothers in Karnataka, thereby identifying the most-at-risk populations (MARP) within the general population. |
| Scheidell 2018 | Haiti | Quantitative | Observational study | The study sought to address gaps in the literature by examining socioeconomic factors and STI/ HIV-related sexual risk behaviours and infection in a sample of 200 pregnant Haitian women receiving antenatal care in Gressier, Haiti, between August and November 2013. |
| Shahid 2022 | Pakistan | Quantitative | Observational study | The study examined the impact of anxiety proneness, marital satisfaction, and perceived social support on fear of childbirth and development of depression among pregnant women in Pakistan. |
| Storeng 2013 | Burkina Faso | Qualitative (Case report) | - | To examine the concept of vulnerability in the context of maternal morbidity and mortality in Burkina Faso, an impoverished country in West Africa. |
| Theophilo 2018 | Brazil | Quantitative | Observational study | To analyse ethnic/racial differences in prenatal and childbirth care. |
| Thomson 2017 | SSA | Quantitative | Mixed methods | The objective of the study was to assess empirical evidence of effects to child and maternal health resulting from structural adjustment administered by the IMF, World Bank and African Development Bank (AfDB). |
| Torres-Torres 2022 | Mexico | Quantitative | Observational study | The study investigated the association of comorbidities and socioeconomic determinants with COVID-19-related mortality and severe disease in pregnant women in Mexico. |

*(Continued)*

**Table 1.** (Continued)

| Author & year | Country | Research method | Study design | Study objectives |
|---|---|---|---|---|
| Webb 2021 | Zimbabwe | Qualitative | Observational study | The study explored the reasons for and experiences of home delivery among women living in rural Zimbabwe. |

1. Individual vulnerability refers to the degree and quality of information available to an individual and its elaboration in his/her practical life.

2. Social vulnerability refers to a set of social factors that influence decision making and/or access to information, services, policies, and actions.

3. Programmatic vulnerability corresponds to the programs and policies designed and implemented by the government and other institutions to respond to certain problems.

**The use of validated scales for vulnerability in maternal health.** Five studies reported on some form of indices and scales used to measure vulnerability in maternal health (Table 2). Only one study used validated indices [38]. The studies used composite indicators such as maternal and child mortality [25], psychosocial risk, disability, overcrowding, dependency, social security, living conditions, and labour and economic precariousness [33], and socioeconomic characteristics as proxy measures of socioeconomic vulnerability. These characteristics include educational attainment, household poverty, employment and ownership of assets [35], baseline distress, childhood abuse history, and HIV diagnosis.

## Vulnerabilities in new-born and child health

Of the included studies, 40 (66%) were publications on vulnerability in the context of child health. Four studies focused on both maternal and child health [47–50]. Most of the studies focused on a single country, while six focused on multiple countries. When reported, the studies were found to be from 40 countries with nine studies from Brazil [48, 51–58], three each from Kenya [58–60], Nigeria [61–63], Democratic Republic of Congo [61, 64, 65], and Uganda [39, 64, 66] and two each from Malawi [61, 67] and Bangladesh [68, 69]. **Fig 3** provides a summary statistic of the studies on vulnerabilities in new-born and child health, while additional information on the characteristics of the studies can be found in online S1 Table.

Most studies were quantitative and observational in design [48, 50–54, 56, 57, 59, 61–64, 68, 70–82], two used quantitative methods and implementation science in design [67, 83], six used qualitative study design [39, 49, 58, 60, 69, 84], and one study was a mixed-method research [66]. Studies on vulnerability in child health have examined the linkages between children's vulnerability, infant mortality [48, 62], sudden infant death [56], acute respiratory infection symptoms [61, 71], poor health outcomes [70, 75], nutritional status [53, 57], and premature birth outcomes [55]. Drachler and colleagues, in their study, developed and validated a social vulnerability index (SVI) [52]. Tudge and colleagues compared the heterogeneity of young children's vulnerability experiences in Kenya and Brazil [58]. Zakayo et al. examined vulnerability in treatment-seeking journeys for acutely ill children [60]. Desclaux and colleagues analysed the 'emergence of a situation of economic vulnerability generated by a public health policy that is nevertheless globally appropriate in terms of accessibility and epidemiological effectiveness' [49].

**Definition of vulnerability in the context of new-born and child health.** Generally, vulnerability was defined in terms of risk, risk exposure, and adaptive behaviours [76]. Kalibala et al. collated examples used by policymakers and program implementers in defining a

**Table 2. Definitions and indices of vulnerability in maternal health.**

| Author & year | Context | How vulnerability was defined or used in article | Vulnerability indices & scales |
|---|---|---|---|
| Arcos 2011 | Maternal health | Vulnerability is defined as any biological and psychological characteristics, social and environmental conditions that affects the structure and functionality of the family and the territory where they live. | The vulnerable families of pregnant women showed a more adverse situation with respect to psychosocial risk, disability, overcrowding, dependency, social security, living conditions, and labor and economic precariousness. |
| Arps 2012 | Maternal health | - | Two types of vulnerability were identified in this study: biological and social vulnerability |
| Den Hollander 2018 | Maternal health | Vulnerability is defined as the likelihood of being 'wronged, that is–being denied adequate satisfaction of certain legitimate claims to physical integrity, autonomy, freedom, social provision, impartial quality of government, social bases of self-respect or communal belonging'. | - |
|  |  | The study points out potential types of vulnerability, which can be categorized according to the analysis of Kipnis. |  |
|  |  | *First, there is cognitive vulnerability, not so much as the mental capacity to deliberate the decision, but because of unfamiliarity with the language and concept of research, risk assessment in medical decision-making, limited education, and health literacy. |  |
|  |  | *Second, there is seemingly deferential vulnerability because of customary obedience to the medical authority. |  |
| Dias 2020 | Maternal health |  | Vulnerability was conceptualized in terms of socioeconomic status such as residing in a municipality characterized by poverty (per capital monthly income less than the Brazilian average income). |
| Ferreira 2012 | Maternal health | - | Vulnerability was conceptualized using indicator of social inequity, distinguished in three interdependent dimensions: individual, social and programmatic. |
| Kaye, 2014 | Maternal health | Vulnerability is defined as exposure to increased health risk and health demands in the context of limited social and economic resources needed for protection against risks or mitigation of resultant social and economic consequences. |  |
| Lavender 2020 | Maternal health | The authors defined vulnerability in terms of three attributes linked to it: social threats to successful transfer, barriers to timely intrapartum care and reparative interventions. | - |
| McNaughton Reyes 2020 | Maternal health | Vulnerability was not defined by the authors | Baseline distress, childhood abuse history, HIV diagnosis |
| Munoz 2013 | Maternal health | According to the authors vulnerability can be defined as "the insecurity and helplessness experienced by communities, families and individuals in their lives as a result of the impact caused by some kind of traumatic social economic event", on one hand, and "the management of resources and strategies used by communities, families and individuals to deal with the effects of that event" on the other hand | - |
| Prates 2008 | Maternal health | Vulnerability was not explicitly defined in this study but was used to describe multiparity [having many children] in the context poverty especially in a context where gender inequality undermines the prospects of young girls and women |  |
| Premji 2020 | Maternal health | Vulnerability is defined in relation to risk of comorbid anxiety and depression in pregnancy. | - |
| Santhakumar 2020 | Maternal health | Vulnerability was not specifically defined by the authors. |  |
| Scheidell 2018 | Maternal health | - | The authors used socioeconomic characteristics as proxy measures of socioeconomic vulnerability. These characteristics comprise of educational attainment, household poverty, employment, and ownership of assets. |

*(Continued)*

**Table 2.** (Continued)

| Author & year | Context | How vulnerability was defined or used in article | Vulnerability indices & scales |
|---|---|---|---|
| Theophilo 2018 | Maternal health | According to the authors defined vulnerability in in terms of: | - |
| | | Individual vulnerability which refers to the degree and quality of information available to the individual and its elaboration in his/her practical life. | |
| | | Social vulnerability refers to the set of social factors that influences decision making and/or access to information, services, policies, and actions. | |
| | | Programmatic vulnerability corresponds to the programs and policies designed and implemented by the government and other institutions to respond to certain problems. | |
| Torres-Torres 2022 | Maternal Health | Not defined but used in the following context. The social vulnerability index provides a summary of four social deficiencies monitored by the National Council for Evaluation of Social Development Policy: educational lag, access to health services, access to essential services in housing and quality of and space in housing. | Using the Dalenius–Hodges stratification method, the social vulnerability index in Mexico may be divided into five categories: very high vulnerability, high vulnerability, medium vulnerability, low vulnerability and very low vulnerability. |

vulnerable child, and are summarised in **Box 1** [66]. Nineteen studies defined vulnerability in the context of new-born and/or child health (**online S2 Table**). The definitions used can be broadly categorised into three categories: socioeconomic, biological, and environmental.

*Biological.* Twelve studies used biological vulnerability in the context of child health. Chiao et al. defined vulnerable children as those who had experienced the death of a family member who had been ill for at least three months during the past 12 months or who were a member of a household with a member who had been ill for at least 3 months during the past 12 months. Davis defined vulnerability as the condition that exists in 'children less than 5 years-old,

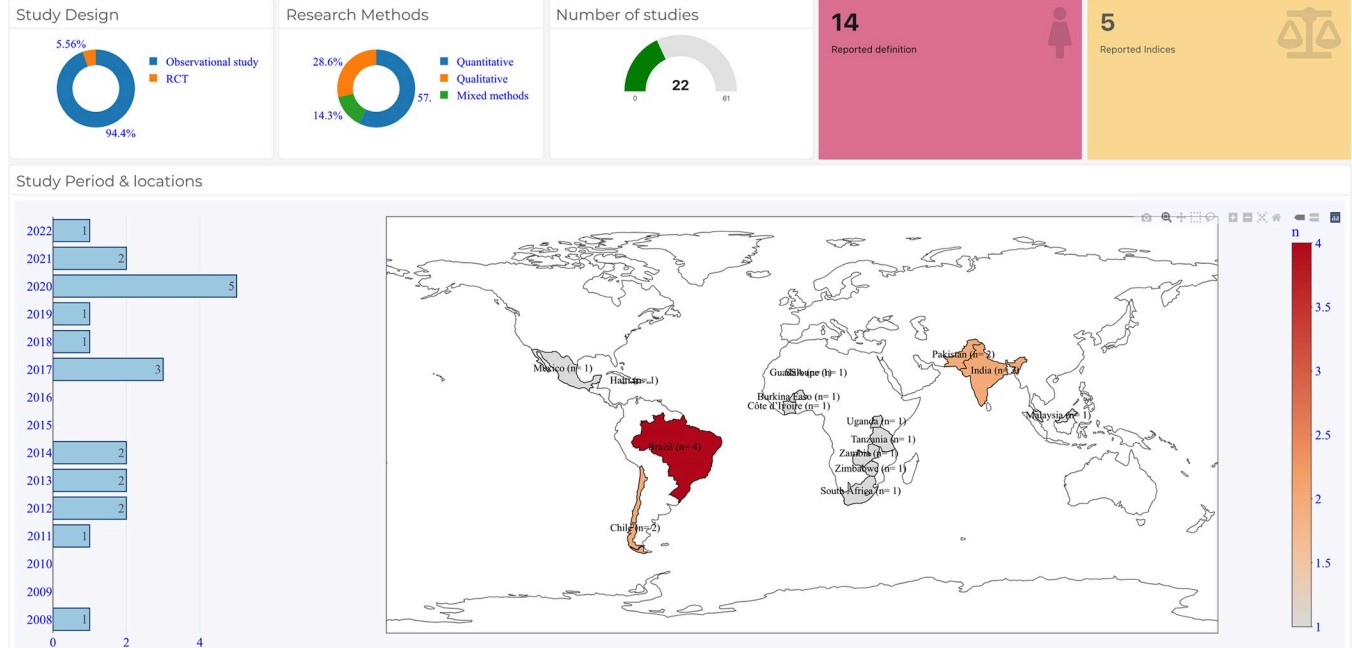

**Fig 2. Sources and distribution of published research on vulnerabilities in maternal health.** Source: Authors generated figure (including map) using R software.

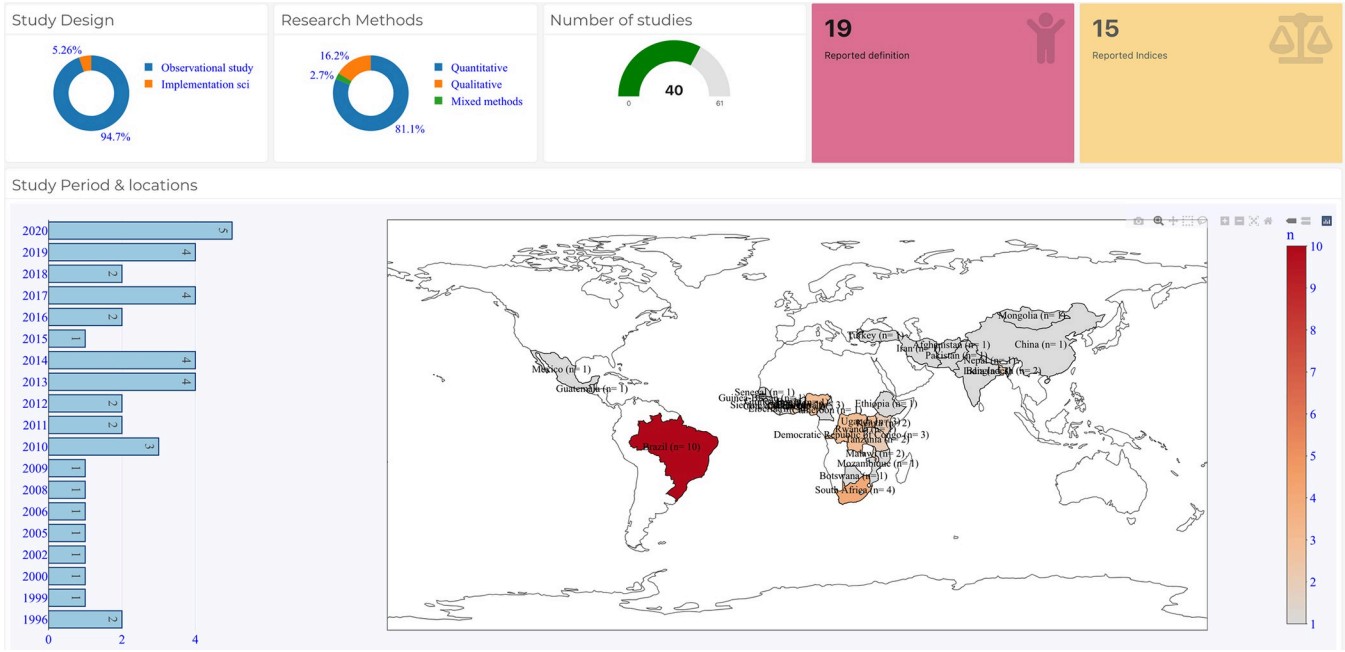

**Fig 3. Sources and distribution of published research on vulnerabilities in new-born and child health.** Source: Authors generated figure (including map) using R software.

pregnant and lactating women, older adults, orphans, the chronically ill and infirm, and the mentally ill' [64]. Siekman noted that 'the concept of vulnerability encompasses both external exposure to risk factors and internal means for coping with those risks without damaging loss. Young children are often considered vulnerable since they are more likely to be harmed by these stressors than others in the general population.' Other studies have conceptualised vulnerability in child health in terms of maternal HIV infection [59], childhood mortality, life-threatening illnesses [72], maternal mortality or death of a family member [61, 78], and pre-term delivery, or low-birth weight [53, 55, 68, 84], Pollitt et al. used a timing approach to define vulnerability, that during periods of rapid brain growth ('brain growth spurt'), the brain is particularly vulnerable to malnutrition.

Box 1. Dimensions of vulnerability *(Kalibala et al. [66]).*

| Level of indicator | Domain | Number of indicators |
|---|---|---|
| Child | Parental status | 8 |
| Child | School attendance | 7 |
| Child | Health and nutrition | 4 |
| Child | Disabilities | 4 |
| Child | Basic material needs | 3 |
| Child | Risk-taking behaviours | 8 |
| Household | Household relationships and situation | 11 |
| Household | Household characteristics | 4 |

*Socioeconomic.* Seven studies used financial vulnerability in the context of child health [39, 47, 49, 57, 60, 62, 70]. Yuan et al. defined vulnerability as the 'combination of multiple factors, including fragile economic foundation, extreme poverty, poor market accessibility, stagnant maternal education level, rapidly inflated population, and serious armed conflict' [62]. Kaye et al. defined vulnerability 'as exposure to increased health risk and health demands in the context of limited social and economic resources needed for protection against risks or mitigation of resultant social and economic consequences' [39]. Zakayo and colleagues stated that vulnerability 'encompasses scarce earning opportunities, impoverished formal education, seasonal drought and food shortages, highly resource-constrained public-sector health services, and strong gender inequities' [60]. Other studies have conceptualised vulnerability in terms of a situation in which a mother is young and unable to afford formula milk to avoid mother-to-child transmission of HIV [47], household wealth status, education levels of caregivers, living arrangements, relationship to head of household [70], socioeconomic vulnerability [49], and social vulnerability of families as related to structural poverty, aggravated by economic problems [57].

*Environmental.* Four studies used environmental vulnerability in the context of child health [8, 50, 69, 77]. Studies have conceptualised vulnerability in terms of severe winter (dzud), which leads to the loss of lives and livestock [69, 77], and Kruk as 'an up-to-date map of human, physical, and information assets that highlight areas of strength and vulnerability' [50, 69, 77], and exposure to intense conflict [8, 50].

**The use of validated scales for vulnerability in new-born and child health.** Fifteen studies reported on scales used to measure vulnerability in child health **(online S2 Table)** [39, 53, 55, 57, 61, 62, 67–70, 76, 77, 83–85]. Childhood vulnerability indices were reported in nine studies [52, 61, 62, 64, 66, 71, 75, 76, 78]. Yuan et al. used the Poverty and Adaptive Capacity Index (PACI) and Population Exposure Index (PEI). Two studies used validated scales to measure new-born and child vulnerability [72, 82]. While Dogan et al. used the Child Vulnerability Scale, Wang et al. used the Devereux Centre for Resilient Children Assessment tools (DECA) scale in child health [39, 53, 55, 57, 62, 67, 68–71, 76, 77, 83–85].

Chiao et al. developed a bespoke community vulnerability index using the following: (1) Community OVC: Average prevalence of OVC aged under 18 years within a community cluster and (2) community rate of sexual violence by male intimate partners: average prevalence of lifetime sexual violence by male partners against women aged 15–49 years within a community cluster [61]. Lara-Valencia et al. used a composite built environment vulnerability index (BEVI) aggregated by area unit which was derived to characterise the spatial pattern of contextual factors to measure vulnerability in child health [76].

Other proxies used to measure vulnerability in child health include:

- Family context, prevalence of stunting [53];

- Infant mortality rate and hospitalisation for conditions treatable by primary care [52];

- Number of people in the household who were working, employment situation of the mother and the head of the household at the time of the child's death, access to health services: access to prenatal care [48];

- Major risk factors for childhood mortality were mother's death (especially due to HIV and tuberculosis), a greater number of children under five years living in the same household and winter season [78];

- A case of socioeconomic vulnerability: A situation in which they have taken on debt with no certainty regarding their future capacity for repayment, which has altered their status and social relationships [49];

- Crude mortality rate [64].

Kalibala et al. examined the dimensions of vulnerability among children using 49 indicators (Box 1) that were grouped into eight domains [66]:

Unlike child vulnerability, maternal vulnerability is under-researched in LMIC settings. Two studies used indices of indebtedness and maternal employment situation to identify maternal vulnerability [47–49], and only one study defined a vulnerable mother as a young mother who is unable to afford formula feeds for her HIV-exposed infant [47].

## Proposing definitions of vulnerability in maternal, new-born, and child health

Taking cues from the included studies, vulnerability has been defined in terms of risk and risk exposure and adaptive behaviours. Three broad categories of socioeconomic, biological, and environmental vulnerability factors were identified. Furthermore, the definitions in the included studies centred around the three fundamental aspects of dependence:

1. Material aspects—money, food, clothing, shelter, health care, and education

2. Emotional aspects—care, love, support, space to grieve, and containment of emotions.

3. Social aspects—absence of a supportive peer group, role models to follow, guidance in difficult situations, and risks in the immediate environment. Finally, the word cloud of constructs was applied to identify keywords in generating new definitions of vulnerability in the MNCH context, as shown in **Fig 4**.

Thus, we propose the following definitions of vulnerability in maternal, new-born, and child health:

1. Women during pregnancy, childbirth, or puerperium and children 5 years old or younger, (9 years according to WHO) who have an increased risk or susceptibility to adverse health outcomes because they experienced limited basic rights or limited resources.

2. Women during pregnancy, childbirth, or puerperium and children, aged 5 years or younger (9 years according to WHO) are at greater risk of experiencing physical, emotional, or poor outcomes because of exposure to one or more adverse factors in their lives.

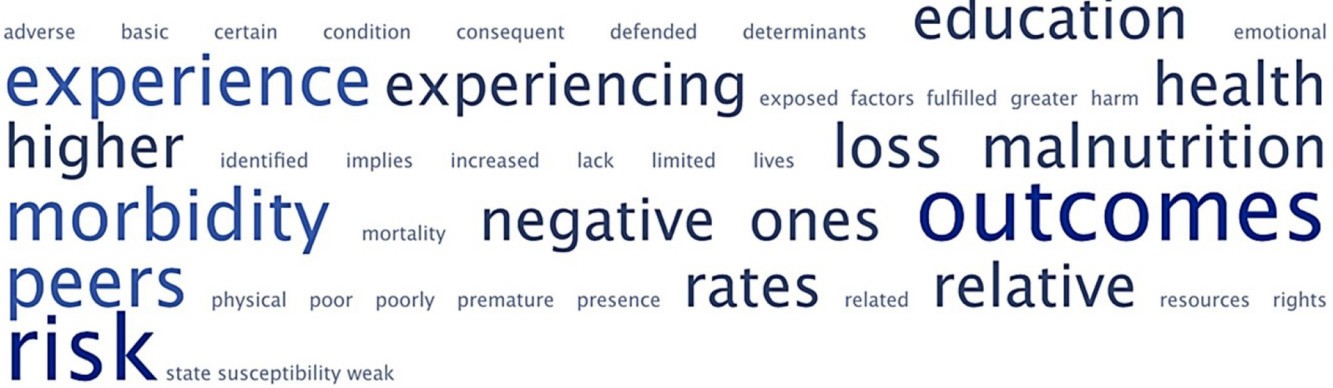

**Fig 4. Word cloud of constructs for defining vulnerability.** Source: Authors generated figure.

3. Women during pregnancy, childbirth, or puerperium and children aged 5 years or younger (9 years according to WHO) who are prone to negative outcomes at a higher rate than their peers due to circumstances beyond their control.

4. The state or condition of women during pregnancy, childbirth, or puerperium and children 5 years-old or younger (9 years according to WHO), who lack health or are susceptible to poor outcomes related to the presence of at least one adverse determinant or lack of access to basic rights and resources.

5. The state or condition of women being weak or poorly defended during pregnancy, childbirth, or puerperium and children 5 years-old or younger (9 years according to WHO), who suffer due to the presence of at least one adverse determinant or lack of access to basic rights and resources.

In choosing a unanimous definition of vulnerability in maternal and child health, we invited seven researchers in this field to rank the above definitions. The results of their ranking with an average ranking score across the reviewers are presented in S3 Table.

Overall, the average ranking score across all the definitions was lowest for the fourth suggested definition of vulnerability in MNCH (S3 Table):

*"State or condition of women in pregnancy, childbirth, or puerperium or children 5 years-old or younger, who lack health or are susceptible to poor outcomes related to the presence of at least one adverse determinant or lack of access to basic rights and resources."*

## Discussion

### Main findings

The aim of this scoping analysis was to identify and synthesise existing definitions and indices of vulnerability in MNCH research and health programs and to use this knowledge to generate a harmonised definition of vulnerability in MNCH. Policymakers and donor agencies are currently engaging in a variety of programs and policy options to enhance MNCH outcomes. It is difficult to design public health interventions that could lead to significant improvements in MNCH without an objective knowledge of concepts and implementations of vulnerability in resource-limited settings.

Women and new-borns are particularly vulnerable during and immediately after childbirth [86]. Every year, an estimated 2.8 million pregnant women and new-borns die, amounting to one death every 11 seconds, from preventable causes [86, 87]. To ensure that women and new-borns are safe before, during, and after childbirth, there is a need to support the development, testing, and scaling-up of innovative solutions to address the underlying vulnerabilities that lead to poor health. A 'vulnerability journey plan' has the potential to recognise how reparative interventions can improve children's and women's capacity for resilience and decrease the level of vulnerability experienced or the outcomes that could be influenced by the vulnerability [88]. The proposed definition can help to identify vulnerable women and children for interventions, and assist policy direction and practice.

The literal description of 'vulnerability' is 'the state or condition of being vulnerable or poorly defended'. Maternal vulnerability was examined in different ways by the various papers identified in this scoping review, suggesting different stakeholders have a different perspective and view to what constitutes vulnerability in maternal health. These were around social, economic, and environmental indices. This has implications on how uniformity in national, regional, and global programs can be effectively described. The limited use of validated indices

and scales in the investigation of maternal vulnerability further highlighted limited interest in this aspect of shared understanding of what maternal vulnerability means across countries. This can be an area for investment and further effort as it would help in the standardization of programs targeted at addressing maternal vulnerability. An examination of studies from Brazil which had the largest number of publications on maternal vulnerability highlights the diverse perspectives of researchers on maternal vulnerability even within the same country.

The definition of vulnerability in relation to children means that they are more susceptible to threats than other family members, community members, or peers [3]. They may be vulnerable to deprivation (lack of food, education, and parental care), exploitation, harassment, neglect, aggression, and HIV infection. Adaptation to vulnerability can vary from resilience to absolute helplessness [3]. While the definition of child vulnerability is frequently mentioned in child development and children's rights literature, it is neither well-described nor well-examined [89–91]. Child vulnerability has been stated to be the result of a variety of individual and environmental factors interacting dynamically over time. In the World Bank's 'Orphans and Vulnerable Children (OVC)' toolkit, vulnerability is described as 'the community of children who experience negative outcomes, such as lack of education, morbidity, and malnutrition, at higher rates than their peers' [3]. It is important to remember, however, that many disadvantaged children fall into more than one category.

Individual factors that contribute to child vulnerability include cognitive, emotional, and physical skills, as well as personal circumstances such as age, disability, a child's personality, or mental health difficulties [92]. Vulnerability factors may be fixed, such as belonging to an ethnic group or having an immigrant status, or modifiable situations such as witnessing maltreatment, becoming an unaccompanied child, or being put in out-of-home care [92]. Environmental factors that contribute to child vulnerability exist at both the family and community levels [92]. Income insecurity and material deprivation, parents' health and health behaviours, parents' education level, family tension, and exposure to intimate partner abuse are family factors. Furthermore, school and neighbourhood cultures are influenced by community factors [92]. Environmental factors highlight the intergenerational nature of child insecurity, as well as the accumulation of vulnerable children within specific families and communities. Acknowledging parents as the primary social shield for children and young adults, the absence of either one of the parents or orphanhood is one of the key determinants of vulnerability among children [3]. Skinner et al. conducted a comprehensive study to define children's 'vulnerability' in Africa and described vulnerable children as 'those who do not have certain of their fundamental rights fulfilled' [93].

Less attention has been paid to the maternal vulnerability portion of MNCH [3]. This scoping review uncovered more research on child health (42 studies) compared to maternal health (18 studies). To fill this research gap, we reviewed the current definitions in the literature and introduced a coherent and detailed definition for both maternal and child vulnerability. We also noted the complexity of the definition of vulnerability in MNCH, as the word 'vulnerability' often has ambiguous applications. Vulnerability is thus a difficult term to define, as shown by the included studies. Moreover, synthesising definitions of vulnerability in published studies, as intended for this project, was difficult.

## Implications for MNCH research and interventions

The population of women of reproductive age and their children in LMICs is substantial and growing. Although the majority of this population is vulnerable, there is a lack of thorough understanding of the socio-demographic factors to leverage the reduction of their vulnerability [94]. A demand-side approach using the viewpoints, goals, and experiences of individuals,

families, and neighbourhoods can provide a deeper understanding of treatment-seeking journeys and identify possible intervention points [95, 96]. Further action is needed that takes into account the following points: (1) the individual level, where the approach emphasises that poorer circumstances and adverse childhood experiences in a person's early life do not inevitably lead to poorer opportunities and outcomes later in life, but instead place children at increased risk of disadvantage; (2) family and other care settings that include a safe and secure environment are important protective factors; and (3) a community-based public health approach to prevention that emphasises the role of the community in developing environments where the best conditions for children to succeed are in place.

## Strengths and limitations of the study

A key strength of this review is the rigorous and transparent search strategy employed to identify the existing literature on reported vulnerability in the context of MNCH. Additionally, we did not limit our search to any language, publication date, or document form. This scoping analysis enabled us to examine how vulnerability has been defined in various studies, as well as indices and scales that have been used in vulnerability programs and research in MNCH in LMICs, which were then used to develop a definition of vulnerability in MNCH. More so, scoping reviews can also serve as an effective prelude to systematic reviews by confirming the relevance of inclusion criteria and prospective topics [97].

Notwithstanding the painstaking approach, this study has some limitations. First, scoping reviews do not evaluate research quality and thus consider data derived from both poor and strong studies. However, the quality of the included studies has no bearing on definitions, scales, or indices of vulnerability presented in published studies, which were both quantitative and qualitative in design. Second, despite our rigorous search strategy, there is a possibility we missed some eligible publications because of the wide scope of vulnerability as an area of research, especially those available as grey literature.

## Conclusion

Adopting a public health approach for addressing vulnerability in maternal and child health in low- and middle-income countries has the potential to dramatically reduce inequality and enhance health and welfare outcomes for the most disadvantaged women and children. To achieve this, interventions should be contextual and address social determinants of health that contribute to vulnerability. The suggested approach calls for multisectoral partnership to implement l interventions critical to minimising vulnerability. It is also important that interventions are early enough to mitigate the negative impact over the life course.

## Supporting information

**S1 Table. Characteristics of included studies on vulnerabilities in newborn and child health.**
(PDF)

**S2 Table. Definitions and indices of vulnerability in new-born and child health.**
(PDF)

**S3 Table. Ranking of suggested definitions of vulnerability in maternal and child health.**
(PDF)

**S1 Fig. Database search strategies.**
(PDF)

## Acknowledgments

We would like to acknowledge Ms. Claire-Helene Mershon, the Program Officer at the Foundation, who guided us through this piece of work. We also appreciate the expert reviewers who participated in ranking the different definitions proposed.

## Author Contributions

**Conceptualization:** Olusesan Ayodeji Makinde.

**Data curation:** Olusesan Ayodeji Makinde, Olalekan A. Uthman, Ifeanyi C. Mgbachi, Nchelem Kokomma Ichegbo, Fatima Abdulaziz Sule, Emmanuel O. Olamijuwon, Babasola O. Okusanya.

**Formal analysis:** Olalekan A. Uthman, Ifeanyi C. Mgbachi, Nchelem Kokomma Ichegbo, Fatima Abdulaziz Sule, Emmanuel O. Olamijuwon.

**Funding acquisition:** Olusesan Ayodeji Makinde.

**Methodology:** Olusesan Ayodeji Makinde, Olalekan A. Uthman, Ifeanyi C. Mgbachi, Babasola O. Okusanya.

**Project administration:** Olusesan Ayodeji Makinde.

**Writing – original draft:** Olusesan Ayodeji Makinde, Olalekan A. Uthman, Ifeanyi C. Mgbachi, Nchelem Kokomma Ichegbo, Fatima Abdulaziz Sule, Emmanuel O. Olamijuwon, Babasola O. Okusanya.

**Writing – review & editing:** Olusesan Ayodeji Makinde, Olalekan A. Uthman, Ifeanyi C. Mgbachi, Nchelem Kokomma Ichegbo, Fatima Abdulaziz Sule, Emmanuel O. Olamijuwon, Babasola O. Okusanya.

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
