## [Decision Letter · Decision Letter 0]

16 Aug 2022

PONE-D-22-08032Vulnerability in Maternal, New-born, and Child Health in Low- and Middle-Income Countries: Findings from a scoping reviewPLOS ONE

Dear Dr. Makinde,

Thank you for submitting your manuscript to PLOS ONE. After careful consideration, we feel that it has merit but does not fully meet PLOS ONE’s publication criteria as it currently stands. Therefore, we invite you to submit a revised version of the manuscript that addresses the points raised during the review process.

The manuscript has been evaluated by two reviewers, and their comments are available below.

The reviewers have raised a number of concerns that need attention. They request additional information on methodological aspects of the study, address limitations of the study, and the inclusion a flow diagram 

Could you please revise the manuscript to carefully address the concerns raised?

We look forward to receiving your revised manuscript.

Kind regards,

Katrien Janin

Staff Editor

PLOS ONE

Journal Requirements:

2. We note that Figures 2 and 3 in your submission contain map images which may be copyrighted. All PLOS content is published under the Creative Commons Attribution License (CC BY 4.0), which means that the manuscript, images, and Supporting Information files will be freely available online, and any third party is permitted to access, download, copy, distribute, and use these materials in any way, even commercially, with proper attribution. For these reasons, we cannot publish previously copyrighted maps or satellite images created using proprietary data, such as Google software (Google Maps, Street View, and Earth). For more information, see our copyright guidelines: http://journals.plos.org/plosone/s/licenses-and-copyright.

a. You may seek permission from the original copyright holder of Figures 2 and 3 to publish the content specifically under the CC BY 4.0 license.  

Reviewers' comments:

Reviewer's Responses to Questions

**Comments to the Author**

1. Is the manuscript technically sound, and do the data support the conclusions?

Reviewer #1: Yes

Reviewer #2: Yes

2. Has the statistical analysis been performed appropriately and rigorously? 

Reviewer #1: N/A

Reviewer #2: N/A

3. Have the authors made all data underlying the findings in their manuscript fully available?

Reviewer #1: Yes

Reviewer #2: Yes

4. Is the manuscript presented in an intelligible fashion and written in standard English?

Reviewer #1: Yes

Reviewer #2: Yes

5. Review Comments to the Author

Reviewer #1: Thank you for the opportunity to review this scoping review manuscript. I think this article has the potential to provide new insights into the field of MNCH, particularly in the LMICs context. I do have some inputs to help the authors to improve the manuscript:

I wonder if similar scoping or systematic reviews exist to help elaborate the state of the art and/or research gap in the introduction.

It is common that after conducting a review, a follow-up scan for articles published since articles were gathered is conducted to make sure no new articles had been published in that time. The one-year gap from 2021 to 2022 should be closed by a quick scan. The authors should do this, and if it was done, detail it in the methods section and as a study limitation.

What are the credentials of the two reviewers?

Placing the context and settings of the selected articles in the discussion section should improve the analysis. For example, comparing definitions and indices of vulnerability in Africa and comparing them to other regions.

I think the authors should concentrate on discussing the limitation of the study, instead of the limitation of the method itself.

Reviewer #2: This study is a scoping review that follows a systematic review process based on a framework and guidelines similar to Prisma. It has a registered protocol.

The number of databases consulted is sufficient and wide-ranging. The gray literature has been reviewed. However, the date of the last search is old (it took place more than 15 months ago). On the other hand, it is striking that it is in March. I wonder if there is some objective reason for this. I suggest updating the search to at least December 31, 2021 or even later.

The type of review and the methodology used really help to address the objective of the study.

Minor comments:

This phrase-ending “, with researchers examining their vulnerabilities.” can be omitted I think.

Can you provide a reference for World Bank Country and Lending groups?

Please, provide a reference for Ovid Northern Light Conference Abstracts instead of the description of the text.

Could you please indicate the first and last name acronyms of the authors (within the manuscript) who participated in the screening and extraction phases? What was the inter-rather percentage agreement between the two literature reviewers? Please, add this proportion in the text.

This phrase is repeated and therefore can be omitted > “Two authors independently charted key information from the included Publications”

By curiosity, what is the utility of using Airtable if Excel is also used?

Table 2. Column “Vulnerability scale” could be deleted. This information may be reflected in the text since no study used it to measure vulnerability.

AUTHORS’ CONTRIBUTIONS, COMPETING INTEREST, ACKNOWLEDGEMENT and FUNDING sections are written in different font format.

I missed a flow diagram. It would be convenient to include it.

38 did not meet the inclusion criteria and were subsequently excluded. Please, indicate exactly how the studies were grouped for each exclusion criterion.

How many years did the search period cover? Please, provide this information into the manuscript.

Language:

This article has been written in an excellent English.

6. PLOS authors have the option to publish the peer review history of their article (what does this mean?). If published, this will include your full peer review and any attached files.

Reviewer #1: **Yes: **Indra Yohanes Kiling

Reviewer #2: **Yes: **Maria Calatrava

---

## [Author Response · Author response to Decision Letter 0]

24 Sep 2022

Dear Editro,

We thank you for the feedback on our work ‘Vulnerability in Maternal, New-born, and Child Health in Low- and Middle-Income Countries: Findings from a scoping review’. We were able to use the feedback from the reviewers to improve the manuscript further. A point by point rebuttal is available below. 

We certainly would be available to address any further comments that the reviewers might have.

Kind Regards’

Olusesan Makinde

For the authors. 

PONE-D-22-08032

Vulnerability in Maternal, New-born, and Child Health in Low- and Middle-Income Countries: Findings from a scoping review

PLOS ONE

Journal Requirements:

We have worked on preparing the manuscript in accordance with the style stated at the links. 

2. We note that Figures 2 and 3 in your submission contain map images which may be copyrighted. All PLOS content is published under the Creative Commons Attribution License (CC BY 4.0), which means that the manuscript, images, and Supporting Information files will be freely available online, and any third party is permitted to access, download, copy, distribute, and use these materials in any way, even commercially, with proper attribution. For these reasons, we cannot publish previously copyrighted maps or satellite images created using proprietary data, such as Google software (Google Maps, Street View, and Earth). For more information, see our copyright guidelines: http://journals.plos.org/plosone/s/licenses-and-copyright.

Response: The maps included in this manuscript were generated by the authors using ‘R’. R is an open-source programming language and there is no copyrighted material included in the entire manuscript. 

This has been done. 

5. Review Comments to the Author

Reviewer #1: Thank you for the opportunity to review this scoping review manuscript. I think this article has the potential to provide new insights into the field of MNCH, particularly in the LMICs context. I do have some inputs to help the authors to improve the manuscript:

I wonder if similar scoping or systematic reviews exist to help elaborate the state of the art and/or research gap in the introduction.

As at the time we started this project, no paper like this had been done. However, we have searched again and did not identify a similar work. 

It is common that after conducting a review, a follow-up scan for articles published since articles were gathered is conducted to make sure no new articles had been published in that time. The one-year gap from 2021 to 2022 should be closed by a quick scan. The authors should do this, and if it was done, detail it in the methods section and as a study limitation.

We have updated the paper based on new papers published between our stated search date and August 2022. 

What are the credentials of the two reviewers?

Two sets of reviewers who have expertise in maternal and child health independently reviewed the studies for inclusion. The Covidence software provided an opportunity for multiple people to review. In fact, all project authors contributed to this level of effort. Whenever there was a discrepancy, this was addressed during a weekly meeting of all investigators. 

Placing the context and settings of the selected articles in the discussion section should improve the analysis. For example, comparing definitions and indices of vulnerability in Africa and comparing them to other regions.

We have added information to improve discussion from the definitions across countries on maternal and child health. However, this was difficult as even within countries, the definitions used by authors was not uniform. So geographic comparison was not that easy. 

I think the authors should concentrate on discussing the limitation of the study, instead of the limitation of the method itself.

This section has been improved. 

Reviewer #2: This study is a scoping review that follows a systematic review process based on a framework and guidelines similar to Prisma. It has a registered protocol.

The number of databases consulted is sufficient and wide-ranging. The gray literature has been reviewed. However, the date of the last search is old (it took place more than 15 months ago). On the other hand, it is striking that it is in March. I wonder if there is some objective reason for this. I suggest updating the search to at least December 31, 2021 or even later.

The type of review and the methodology used really help to address the objective of the study.

Minor comments:

This phrase-ending “, with researchers examining their vulnerabilities.” can be omitted I think.

This has been addressed.

Can you provide a reference for World Bank Country and Lending groups?

This has been included.

Please, provide a reference for Ovid Northern Light Conference Abstracts instead of the description of the text.

This has been done.

Could you please indicate the first and last name acronyms of the authors (within the manuscript) who participated in the screening and extraction phases? 

Due to the Covidence software we used for managing the data, all authors participated in the screening and review of the articles. All authors met once every week to deliberate on any discrepancies till this stage of the research was completed. Thus, a statement mentioning the role all authors played has been further included in this section. 

What was the inter-rather percentage agreement between the two literature reviewers? Please, add this proportion in the text.

We didn’t track this proportion during the study. 

This phrase is repeated and therefore can be omitted > “Two authors independently charted key information from the included Publications”

This has been edited accordingly

By curiosity, what is the utility of using Airtable if Excel is also used?

Table 2. Column “Vulnerability scale” could be deleted. This information may be reflected in the text since no study used it to measure vulnerability.

It was for convenience and because we were all working remotely, Airtable was a better option when we started with the large dataset. 

AUTHORS’ CONTRIBUTIONS, COMPETING INTEREST, ACKNOWLEDGEMENT and FUNDING sections are written in different font format.

I missed a flow diagram. It would be convenient to include it.

It was added as Figure 1. 

38 did not meet the inclusion criteria and were subsequently excluded. Please, indicate exactly how the studies were grouped for each exclusion criterion.

We have added information on reasons for exclusion. Due to the update search we did, the number of articles excluded has increased. "Of these, 49 did not meet the inclusion criteria and were subsequently excluded, because of wrong population (n=25), wrong outcomes reported (n=16) and conducted in high-income countries (n=8)."

How many years did the search period cover? Please, provide this information into the manuscript.

There was no limit to the search.

---

## [Decision Letter · Decision Letter 1]

10 Oct 2022

PONE-D-22-08032R1Vulnerability in Maternal, New-born, and Child Health in Low- and Middle-Income Countries: Findings from a scoping reviewPLOS ONE

Dear Dr. Makinde,

Thank you for submitting your manuscript to PLOS ONE. After careful consideration, we feel that it has merit but does not fully meet PLOS ONE’s publication criteria as it currently stands. Therefore, we invite you to submit a revised version of the manuscript that addresses the points raised during the review process.

We look forward to receiving your revised manuscript.

Kind regards,

Chhabi Lal Ranabhat

Academic Editor

PLOS ONE

Journal Requirements:

Additional Editor Comments:

Dear Authors,

An interesting paper particularly in low and middle income countries. I went through your paper and reviewers comments. I have following suggestion to revise and resubmit.

1) Your conclusion writing in abstract is going on wrong way. It does not accept the previous study statements and it does not accept any quote writing "“State or condition of women in ........resources". Please remove it.

2) Add some sentences or one paragraph where is the previous study gap and how this study bridge that gap in scientific community. Here are some reference papers, please read and cite as necessary.

i) https://www.frontiersin.org/articles/10.3389/fpubh.2019.00414/full

ii) https://archpublichealth.biomedcentral.com/articles/10.1186/s13690-019-0331-7

iii)https://www.tandfonline.com/doi/abs/10.1080/03630242.2016.1267689

iv) https://www.nature.com/articles/s41586-019-1545-0

Good luck!!!

Reviewers' comments:

Reviewer's Responses to Questions

**Comments to the Author**

1. If the authors have adequately addressed your comments raised in a previous round of review and you feel that this manuscript is now acceptable for publication, you may indicate that here to bypass the “Comments to the Author” section, enter your conflict of interest statement in the “Confidential to Editor” section, and submit your "Accept" recommendation.

Reviewer #1: All comments have been addressed

Reviewer #2: All comments have been addressed

2. Is the manuscript technically sound, and do the data support the conclusions?

Reviewer #1: Yes

Reviewer #2: Yes

3. Has the statistical analysis been performed appropriately and rigorously? 

Reviewer #1: Yes

Reviewer #2: N/A

4. Have the authors made all data underlying the findings in their manuscript fully available?

Reviewer #1: Yes

Reviewer #2: Yes

5. Is the manuscript presented in an intelligible fashion and written in standard English?

Reviewer #1: Yes

Reviewer #2: Yes

6. Review Comments to the Author

Reviewer #1: Thank you for addressing all of the comments. I think current version of the manuscript is worthy to be published. Congratulations to the authors!

Reviewer #2: Good work!

I have one comment: the figure 1 (flow chart) needs to be updated with new data.

Please, considere it.

7. PLOS authors have the option to publish the peer review history of their article (what does this mean?). If published, this will include your full peer review and any attached files.

Reviewer #1: **Yes: **Indra Yohanes Kiling

Reviewer #2: **Yes: **María Calatrava

---

## [Author Response · Author response to Decision Letter 1]

11 Oct 2022

Dear Editor,

Greetings and thanks for asking for further updates on the manuscript.

We have used the feedback provided to update the manuscript further.

Below, we provide a point-by-point response to your comments.

Thanks for your support. 

Journal Requirements:

The references were reviewed and only appropriate references have been included and all were listed in the bibliography. 

Additional Editor Comments:

Dear Authors,

An interesting paper particularly in low and middle income countries. I went through your paper and reviewers comments. I have following suggestion to revise and resubmit.

1) Your conclusion writing in abstract is going on wrong way. It does not accept the previous study statements and it does not accept any quote writing "“State or condition of women in ........resources". Please remove it.

2) Add some sentences or one paragraph where is the previous study gap and how this study bridge that gap in scientific community. Here are some reference papers, please read and cite as necessary.

i) https://www.frontiersin.org/articles/10.3389/fpubh.2019.00414/full

ii) https://archpublichealth.biomedcentral.com/articles/10.1186/s13690-019-0331-7

iii)https://www.tandfonline.com/doi/abs/10.1080/03630242.2016.1267689

iv) https://www.nature.com/articles/s41586-019-1545-0

Good luck!!!

We have worked on the abstract and rewritten the conclusion as suggested. 

We have reviewed the articles you shared and have incorporated two that were much in line with the topic. 

Reviewer #2: Good work!

I have one comment: the figure 1 (flow chart) needs to be updated with new data.

Please, considere it.

We have replaced Fig 1 with the appropriate version.

---

## [Editor Report · Decision Letter 2]

13 Oct 2022

Vulnerability in Maternal, New-born, and Child Health in Low- and Middle-Income Countries: Findings from a scoping review

PONE-D-22-08032R2

Dear Authors,

We’re pleased to inform you that your manuscript has been judged scientifically suitable for publication and will be formally accepted for publication once it meets all outstanding technical requirements.

Kind regards,

Chhabi Lal Ranabhat

Academic Editor

PLOS ONE
---

## [Editor Report · Acceptance letter]

17 Oct 2022

PONE-D-22-08032R2 

Vulnerability in Maternal, New-born, and Child Health in Low- and Middle-Income Countries:  Findings from a scoping review 

Dear Dr. Makinde:

I'm pleased to inform you that your manuscript has been deemed suitable for publication in PLOS ONE. Congratulations! Your manuscript is now with our production department. 

Kind regards, 

on behalf of

Dr. Chhabi Lal Ranabhat 

Academic Editor

PLOS ONE